# A Cascaded Individual Cow Identification Method Based on DeepOtsu and EfficientNet

Ruihong Zhang [1], Jiangtao Ji [1], Kaixuan Zhao [1,*], Jinjin Wang [1], Meng Zhang [1] and Meijia Wang [2]

1   College of Agricultural Equipment Engineering, Henan University of Science and Technology, Luoyang 471023, China
2   School of Electronic Information and Artificial Intelligence, Shannxi University of Science & Technology, Xi'an 710021, China
*   Correspondence: kx.zhao@haust.edu.cn

**Abstract:** Precision dairy farming technology is widely used to improve the management efficiency and reduce cost in large-scale dairy farms. Machine vision systems are non-contact technologies to obtain individual and behavioral information from animals. However, the accuracy of image-based individual identification of dairy cows is still inadequate, which limits the application of machine vision technologies in large-scale dairy farms. There are three key problems in dairy cattle identification based on images and biometrics: (1) the biometrics of different dairy cattle may be similar; (2) the complex shooting environment leads to the instability of image quality; and (3) for the end-to-end identification method, the identity of each cow corresponds to a pattern, and the increase in the number of cows will lead to a rapid increase in the number of outputs and parameters of the identification model. To solve the above problems, this paper proposes a cascaded dairy individual cow identification method based on DeepOtsu and EfficientNet, which can realize a breakthrough in dairy cow group identification accuracy and speed by binarization and cascaded classification of dairy cow body pattern images. The specific implementation steps of the proposed method are as follows. First, the YOLOX model was used to locate the trunk of the cow in the side-looking walking image to obtain the body pattern image, and then, the DeepOtsu model was used to binarize the body pattern image. After that, primary classification was carried out according to the proportion of black pixels in the binary image; then, for each subcategory obtained by the primary classification, the EfficientNet-B1 model was used for secondary classification to achieve accurate and rapid identification of dairy cows. A total of 11,800 side-looking walking images of 118 cows were used to construct the dataset; and the training set, validation set, and test set were constructed at a ratio of 5:3:2. The test results showed that the binarization segmentation accuracy of the body pattern image is 0.932, and the overall identification accuracy of the individual cow identification method is 0.985. The total processing time of a single image is 0.433 s. The proposed method outperforms the end-to-end dairy individual cow identification method in terms of efficiency and training speed. This study provides a new method for the identification of individual dairy cattle in large-scale dairy farms.

**Keywords:** dairy cow; individual identification; body pattern image; binarization; cascaded classification

## 1. Introduction

With the improvement in people's living standards and consumption levels, the demand for animal protein is gradually increasing [1,2]. With limited environmental resources and increasing labor costs, the large-scale development of farms is the key to meeting the above needs [3]. In the management of large dairy farms, the use of manual methods to monitor the health and production of each cow is not only time-consuming and labor-intensive, but also subjective. Therefore, precise dairy farming technology, which is used to monitor individual dairy cows in real time and make timely management

decisions, is an important way to improve efficiency and reduce costs in large-scale farms. The automatic identification of the individual identity of dairy cows is the premise and foundation for achieving precision management.

Currently, passive radio frequency identification (RFID) technology [4], active RFID technology [5], and other wireless technologies—such as radar [6] and wireless local area networks [7]—are sensor-based individual identification methods commonly used on farms. The above methods generally have high accuracy and wide applicability, but the identification system requires cows to wear ear tags or transponders, which not only cause stress to cows but are also prone to damage or loss [8].

In recent years, with the development of computer vision technology in dairy cow behavior analysis and health monitoring [9–13], individual cow identification methods based on biometrics have become a research hotspot [14,15]. Noncontact individual identification systems based on biometrics have the advantages of low cost and not inducing stress responses in cows and can be integrated into intelligent monitoring systems for cows. The muzzle print [16], iris [17], head contours and textures [18,19], tailhead pattern [20], body pattern [14], and gait characteristics [21] of a cow can be used as distinguishable cow identifiers. However, it is difficult to obtain clear images of specific areas of a cow's head—such as the muzzle print, iris, head texture, etc.—which requires the cow to have a high degree of coordination, and the shooting results are easily affected by the shooting angle and position. Gait activity and characteristics will change due to changes in the physiological state of cows (such as lameness, estrus, etc.), resulting in the reduced accuracy of individual identification. In addition, some scholars have tried to identify individual cows by locating and recognizing numbers on tags worn by cows (e.g., ear tags [22] and collar ID tags [23,24]), but the implementation of tags requires additional manpower and material resources.

Body pattern refers to the regular distribution of black and white hair in the trunk area of Holstein cows. The distribution area of the body pattern is wide, and the body pattern image can be obtained by collecting side-looking, top-looking images or videos of a cow in the walking process. Zhao et al. [14] extracted a $48 \times 48$ matrix from a cow's trunk image as the eigenvalue, and a convolutional neural network was constructed and trained as the individual cow identification model. The dataset contained 30 cows, and 90.55% of the images were correctly identified in the test. Li et al. [20] located a cow's tailhead, and the contour of the black and white pattern of the tailhead was obtained by binary image processing. Then, the feature matrix was extracted, and classification was carried out. The dataset contained 10 cows, and the final accuracy was 99.7%. The number of cows studied by the above methods is small, and the adaptability to large-scale farms is unknown. Therefore, scholars have begun to build datasets containing more cows for the individual identification of cow groups in large-scale farms. He et al. [25] preprocessed the back images of 89 cows and constructed a milking individual cow identification model based on the improved YOLO v3 algorithm, which achieved 95.91% identification accuracy. Hu et al. [8] used YOLO to detect the position of cows and separated the head, body, and legs from the detection frame of cows. The features of these three parts were extracted, fused, and classified. It achieved 98.36% accuracy for 93 cows. Shen et al. [26] used the YOLO model to obtain the detection box containing the cow, and the AlextNet algorithm was fine-tuned to identify cow individuals. The constructed dataset contained 105 cows, and the identification accuracy was 96.65%.

The output end of the individual cow identification model constructed by the method directly corresponds to the number of cows, the increase in the number of output ends causes an increase in identification network parameters, and the time cost for individual identification and retraining of the network correspondingly increases. In addition, the body patterns of different cows may be similar, which will increase the difficulty of correct identification. At the same time, the above methods all use RGB body images as the input of the identification model. However, the farming environment of dairy cows is complex, and light, stains, fences, and so on will affect the quality of body pattern images. This

means that the identification model should not only judge the classification of the target but also eliminate interference in the image, which increases the complexity of the identification network as well.

In view of the above problems, a cascaded dairy individual cow identification method based on EfficientNet [27] and DeepOtsu [28] is proposed in this paper and was applied to large-scale dairy farms. It can realize a breakthrough in dairy cow group identification accuracy and speed by binarization and cascaded classification of dairy cow body pattern images. The specific implementation steps of the proposed method are as follows: first, the body pattern image is obtained by using YOLOX to locate the trunk region of the cow, and then, the body pattern image is binarized by the DeepOtsu model. Then, primary classification is carried out according to the proportion of black pixels in the binary image. Then, for each subcategory obtained by primary classification, the EfficientNet model is used for secondary classification to identify the identity of the individual cow. Compared with the end-to-end identification method, the proposed cascaded identification method reduces the number of outputs and parameters of the individual identification model, which provides a new idea for the individual identification of dairy cows on large-scale dairy farms.

In general, we proposed a cascaded method for the individual identification of dairy cows that mainly consists of three modules: cow trunk localization, body pattern image binarization and cascaded classification. The main contributions of this paper are as follows.

- A new method of individual cow identification was proposed. The method comprises the following steps. First, the cow trunk region was detected to obtain a body pattern image. Then, the pattern image was binarized to highlight the distribution characteristics of the black and white patterns. Finally, the binary pattern image was classified to identify the individual cow.
- The body pattern images of cows were classed by utilizing a cascaded classification method. The method can reduce the number of output ends of the classification model and improve the efficiency of the training. The identification accuracy, speed, and training time of the proposed method were compared with those of the end-to-end identification method, and the results showed that the proposed method is superior to the end-to-end method.
- The body pattern image was binarized by the deep learning method. The experimental results showed that the deep learning method can better describe the features of RGB body pattern images, remove the interference factors in the images, and achieve better binarization accuracy.

## 2. Materials and Methods

### 2.1. Dataset Construction

2.1.1. Video Acquisition

In this study, 118 lactating Holstein cows were filmed at Coldstream Research Dairy Farm, University of Kentucky, USA. The cows returned to the cowshed after milking. A straight corridor was set on the only way back to the cowshed. Two electric fences were used on both sides as the boundary of the corridor. The width of the corridor was 2 m. The cows passed a weighing device before entering the corridor. The weighing device has electronically controlled doors to ensure an interval between cows when passing through the corridor, so individuals overlapping will not happen in the video. The image acquisition system consisted of a Nikon D5200 camera (Nikon, Tokyo, Japan) and a tripod, which was fixed on one side of the aisle at a distance of 3.5–4 m from the corridor and a height of 1.5 m from the ground. The specific location is shown in Figure 1. The acquisition time of the video was from 16:00 to 18:00 on sunny days from August to October 2016. The camera used a 35 mm lens (Nikon AF-S DX 35 mm f/1.8 G) (Nikon, Tokyo, Japan), and ISO 400, autoexposure and autofocus modes were selected when acquiring images. When a cow passed through the corridor, video shooting began, and when the cow walked to the

right edge of the field of view, shooting ended. The cows were filmed multiple times on different dates.

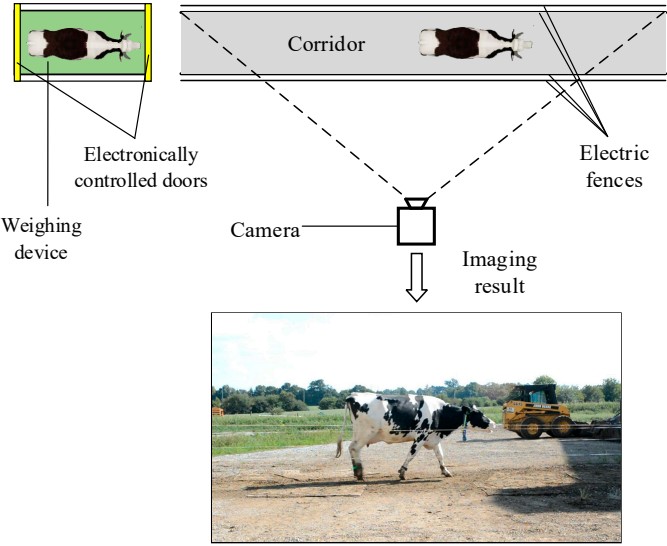

**Figure 1.** Diagram of the video acquisition system.

2.1.2. Video Decomposition and Processing

The collected videos were analyzed, and the overexposed videos were eliminated to obtain side-looking walking videos of cows. The construction of the dataset mainly comprised the following steps: (1) decomposing the video into image frames; (2) selecting the image frames randomly and quantitatively; (3) classifying the images; (4) normalizing the number of images; and (5) constructing and dividing the subdatasets.

(1) Decomposing the video into image frames. Video decomposition technology was used to decompose the cow side-looking walking videos into frame-by-frame images. The resolution of the cow side-looking image was 1280 pixels (horizontal) × 720 pixels (vertical).

(2) Selecting the image frames randomly and quantitatively. For each walking video, 100 side-looking walking images were randomly selected, and it was ensured that each image contained the complete trunk of the cow.

(3) Classifying the images. The side-view walking images belonging to the same cow were classified and placed into a folder.

(4) Normalizing the number of images. For a folder containing more than 100 images, 100 images were randomly selected as the image dataset corresponding to the cow. The final constructed dataset contained 11,800 images of 118 cows.

(5) Constructing and dividing the subdataset. Due to the large number of samples in the dataset, it is labor-intensive and unnecessary to annotate all the images to train and test the model. Therefore, 10 images of each cow in the dataset were randomly selected to construct a subdataset to train and test the trunk detection and body pattern binarization model. The subdataset contained 1180 images of 118 cows, and the subdataset was divided into a training set, validation set, and test set at a ratio of 5:3:2.

2.1.3. Image Annotation

The cascaded individual cow identification method proposed in this paper needs two annotations during training. One annotation involves labeling the trunk region when training the trunk location model, and the other involves labeling the body pattern in image binarization when training the body pattern binarization model. For trunk region annotation, the Labelme image annotation tool was used. For the body pattern image binarization annotation, the 3D drawing tool of the Windows system was used. The above annotation process was only processed for subdatasets.

### 2.1.4. Training and Test Platform

The YOLOX detection model, the DeepOtsu binarization model, and the Efficient-Net classification model were trained and tested on the same hardware and software platform. The CPU of the platform was an Intel (R) Xeon (R) with 8 G memory. The graphics card of the platform was an NVIDIA Tesla K80(NVIDIA, CA, USA) with 12 G memory. The software environment for training and testing was an Ubuntu 18.04 LTS 64-bit system. The programming language was Python 3.8. CUDA11.0 and cuDNN8.0 were used as the parallel computing architecture and GPU acceleration library for deep neural networks, respectively.

### 2.2. Detection of the Trunk Area

The body pattern of a cow is mainly concentrated in the trunk area. To eliminate the influence of an irrelevant environment, the trunk area in the side-looking walking image of a cow was located. Existing methods for cow individual location include the frame difference method [14], Gaussian mixture model [29], and YOLO model based on a convolutional neural network (CNN) [25]. The frame difference method uses the difference operation of the adjacent frame images in the video image sequence to obtain the contour of the moving cow target. The Gaussian mixture model obtains the position of the moving cow target by analyzing the change in the gray value of the pixel point in the video. The above two methods need to analyze the continuous sequence of images in the video, and a moving interference object in the background will greatly affect the detection accuracy of the cow target. The YOLO model [30] is a one-step target detection algorithm based on a CNN that uses convolution to extract the features of the image and directly outputs the location and category of the target according to the features. To detect the trunk region in this study with many external interference factors, it is more appropriate to use the detection model based on deep learning. YOLOX was proposed by Ge et al. [31], and its performance exceeds that of the YOLO series of algorithms. YOLOX achieves 50.0% AP on COCO (1.8% higher than YOLOv5 and 2.5% higher than YOLOv4) [31], and the precision of YOLOX is much higher than that of YOLOv3 (33.0% AP). Therefore, we finally decided to use YOLOX to detect the trunk area of dairy cows.

The YOLOX model was built based on YOLOv5 and mainly included four modules: input, backbone, neck, and prediction modules. The structure of YOLOX is shown in Figure 2. When the image to be detected is input into the network, it is first adjusted to a size of $416 \times 416$ and then sent to the backbone of the network for feature extraction, obtaining three effective feature layers. In the neck module, a series of convolution, upsampling, and downsampling operations and others are carried out on the three effective feature layers to fuse different feature layers and strengthen the feature extraction process. Finally, the prediction module performs a convolution operation on the fused feature layers to obtain the category and position information of the detected target.

After detection, the original image was cropped according to the coordinate information of the detection frame to obtain the body pattern image of the cow. A schematic of the processing method is shown in Figure 2.

The training set in the subdataset was used to train the YOLOX-based trunk detection model. After training, the images of the test set in the subdataset were put into the trained detection model to test its performance. In this paper, AP and $AP^{IoU=0.75}$(AP75) in the index of the COCO dataset were used to evaluate the accuracy of the trunk detection model. These two indicators are defined as follows. The IoU (intersection over union) is a value used to measure the degree of overlap between a prediction box and a groundtruth box, and its formula is

$$\text{IoU} = \frac{S_p \cap S_g}{S_p \cup S_g} \tag{1}$$

where $S_p$ represents the area of the predicted bounding box, and $S_g$ represents the area of the groundtruth bounding box. IoU threshold is used to determine whether the content in the prediction box is a positive sample. For the target detection model, the commonly

used evaluation indices were precision *P* (Precision) and *R* (Recall), and their calculation formulas are

$$P = \frac{TP}{TP + FP} \tag{2}$$

$$R = \frac{TP}{TP + FN} \tag{3}$$

where *TP* represents the number of correctly predicted targets. *FP* represents the number of falsely predicted targets, that is, the background was mistaken for a positive sample. *FN* represents the number of missed targets, that is, a positive sample was mistaken as the background. For each prediction box, a confidence value was generated, indicating the credibility of the prediction box. Different combinations of *P* and *R* were obtained by setting different confidence thresholds. Taking *P* and *R* as vertical and horizontal coordinates, respectively, the PR curve could be drawn. When the IoU threshold was set to 0.75, the area under the PR curve was $AP^{IoU=0.75}$ (AP75). AP was averaged over multiple IoU values. Specifically, we used 10 IoU thresholds of 0.50:0.05:0.95. AP and AP75 would comprehensively reflect the performance of the detection model.

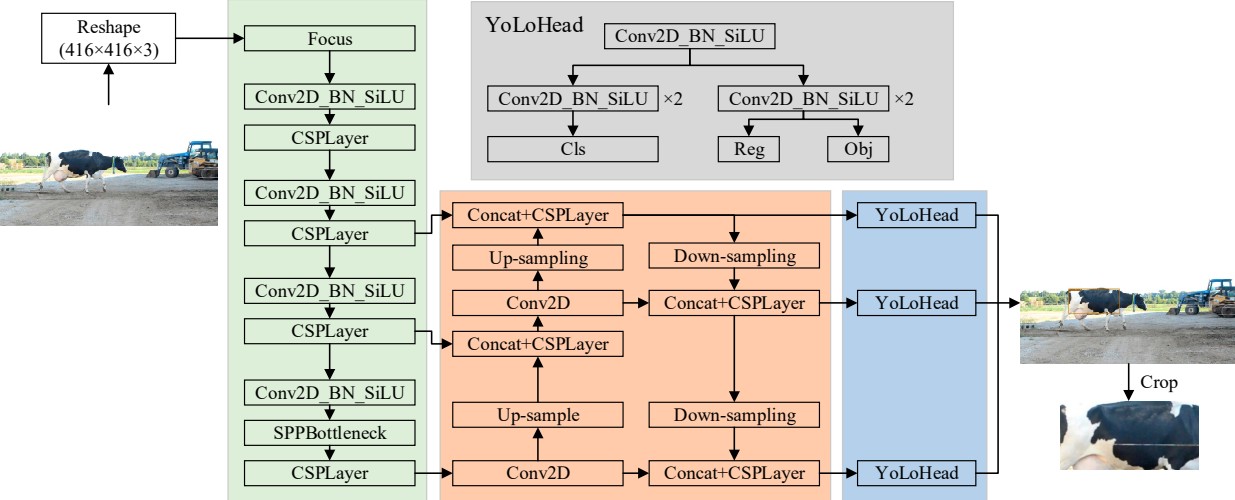

**Figure 2.** Acquisition of the cow body pattern image based on YOLOX.

After testing, the trained YOLOX model was used to detect the trunk areas of the remaining cow side-looking walking images in the dataset to obtain body pattern images of all cows.

### 2.3. Binarization of Body Pattern Images

The most prominent feature in the body pattern image of the trunk region is the distribution of black and white patterns. Therefore, in this study, the distribution of black and white patterns was used as the basis for the classification of body pattern images, that is, the identity of individual cows. To highlight the main feature of black and white patterns in the image, the body pattern image was binarized to make the area where black hair is located black and the area where white hair is located white in the image.

#### 2.3.1. Traditional Binarization Method

In this study, due to the obvious color difference between black hair and white hair, two traditional binarization methods—the Otsu method and the color-based binarization method—were used to segment the cow body pattern images. The Otsu method uses the gray characteristics of the image to divide the image into two parts—foreground and background—and when the difference is greatest, the optimum threshold is taken. When using this method for binarization, the image needs to be processed into a gray image first.

In this paper, the weighted average method (Formula (4)) was used to perform grayscale processing on the image, and then, the Otsu method was used to perform binarization.

$$\text{Gray}(i, j) = (R(i,j) + G(i,j) + B(i,j))/3 \tag{4}$$

where $R(i, j)$, $G(i, j)$, and $B(i, j)$ represent the three components of each pixel point of the color image and Gray (i, j) represents the composite value of the three components, that is, the gray value of each pixel point of the processed gray image.

In addition, according to the statistics of the pixel points in the region where the black and white hairs are located in the cow trunk image, an image binarization method based on color feature was designed, as shown in Figure 3. The method determines whether the pixel point is assigned black (0) or white (1) according to the $R$, $G$, and $B$ values of each pixel point in the image. In Figure 3, $R$, $G$, and $B$ represent the three component values of each pixel point.

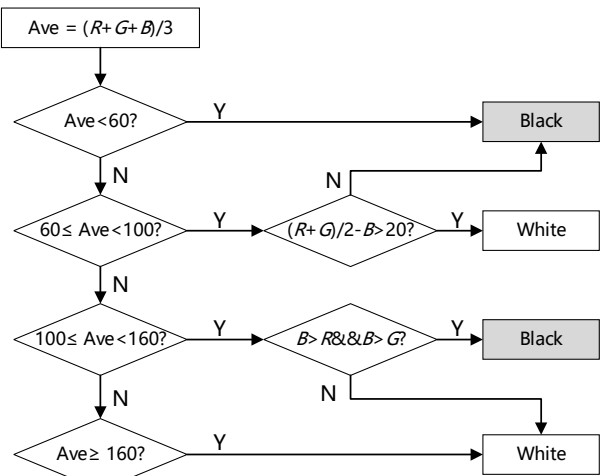

**Figure 3.** Binarization method based on color features.

2.3.2. DeepOtsu

There were noises from light, stains, occlusion in the background of the cow trunk images, which will lead to wrong binarization results. For example, the reflection caused by strong light makes the black hair area very bright, then the binarization result of this black hair area is easily misclassified as white (value 1); the presence of stains in the white hair area will cause the area to darken, and its binarization result is easily misclassified as black (value 0). Therefore, it is necessary to eliminate the background noise in the image in order to achieve better binarization effect. The binarization method based on simple features such as color and gray distributions may not achieve satisfactory results, because these features cannot eliminate these noises well. CNNs can automatically learn rich and useful features from images and have good performance in image segmentation, classification, target detection, and other tasks. Therefore, in this study, a CNN was used to solve the binary segmentation problem of body pattern images. The DeepOtsu model was proposed by He and Schomaker [28] and mainly solves the document enhancement and binarization problem. Unlike the traditional method of predicting each pixel value, the author proposed a model of learning degradation in images. The model processed the degraded images ($x$) into uniform images ($x_u$) using the CNN (Formula (5)), which are noise-free. Then, the images were binarized (Formula (6)) using existing single methods.

$$x_u = \text{CNN}(x) + x \tag{5}$$

$$x_b = \text{B}(x_u) \tag{6}$$

where $x_u$ represents the processed uniform images, $x$ represents the degraded images, B represents an existing binarization method (for example, Otsu), and $x_b$ represents the binarized image.

Because there is also background noise affecting binarization segmentation in the body pattern image, referring to the ideas in the above paper [28], this paper uses U-Net [32] to learn the interference factors in the image and eliminate these negative effects. U-Net is an image segmentation algorithm with a simple convolutional neural network structure, which is also called the encoder–decoder structure. The function of the encoder is to extract the features of different depths of the image, which is realized by convolution and pooling operations. The role of the decoder is to output a segmentation result based on the feature map, which is implemented using upsampling (deconvolution) and feature map concatenation. In the binarization task of cow body pattern images, only two categories are employed, which does not require a very deep or complex network structure. Because the number of images used for training is small, it is easy to cause overfitting by using a large network. U-Net with a simpler structure is sufficient to learn the useful features in cow body pattern images and eliminate noise from the background. The structure of U-Net is shown in Figure 4. After segmentation, a gray image with noise removed is obtained, and then it is processed into a binary image by the Otsu method.

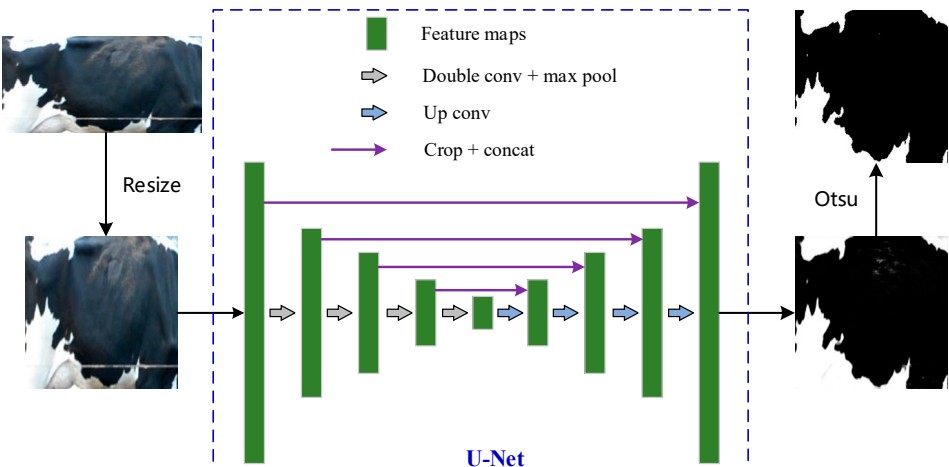

**Figure 4.** Flowchart of the binarization of a cow body pattern image based on DeepOtsu.

Because the sizes of the body pattern images obtained by the detection model are different, size normalization processing was carried out. The size of all the body pattern images was processed to be 256 (pixels) × 256 (pixels) by using a bicubic interpolation method. The subdataset was used to train and test the DeepOtsu model, the $Acc_{seg}$ index was used to evaluate the segmentation accuracy of the model, and the detection time of a single image was used as the index to evaluate the efficiency of the model. $Acc_{seg}$ is calculated with Formula (7):

$$Acc_{seg} = \frac{TP + TN}{TP + TN + FP + FN} \tag{7}$$

where $TP$ represents the number of correctly segmented white pixels, $TN$ represents the number of correctly segmented black pixels, $FP$ represents the number of incorrectly segmented white pixels, and $FN$ represents the number of incorrectly segmented black pixels. In addition, the two traditional binarization methods were used to binarize the cow body pattern images in the test set, and the accuracy index $Acc_{seg}$ was calculated. By comparing the accuracy of the three methods, we can determine which method is used to binarize the cow body pattern images.

After the completion of the comparative experiment, the remaining body trunk images in the dataset were processed with the best binarization model to obtain the binary body pattern images of all cows.

*2.4. Cascaded Classification of Body Pattern Images*

For the end-to-end dairy cow individual automatic identification system, the number of dairy cows corresponds to the number of outputs of the individual identification model, and the number of outputs directly affects the quantum parameter and precision of the identification model. In theory, the more output terminals there are, the lower the efficiency and accuracy of the network. In this paper, a cascaded classification method was proposed to reduce the number of outputs of the individual cow identification network. The specific implementation steps are as follows. First, the image was classified according to the proportion of black pixels in the cow body pattern image to realize primary classification. Then, for each subcategory obtained by primary classification, classification was carried out according to the pattern features to realize secondary classification. The cascaded classification method can reduce the number of network parameters without reducing the accuracy, thus improving the efficiency and accuracy of the individual cow identification network.

2.4.1. Primary Classification

As the dataset processed in this study includes 118 cows, it is reasonable to divide the cows into four categories in primary classification. Classification is based on the *B-pro* value, the proportion of black pixels in the binary body pattern image. The images of *B-pro* falling in the interval [0, 0.25) were classified as category I; the images of *B-pro* falling in the interval [0.25, 0.5) were classified as category II; the images of *B-pro* falling in the interval [0.5, 0.75) were classified as category III; and the images of *B-pro* falling in the interval [0.75, 1) were classified as category IV. The primary classification process is shown in Figure 5.

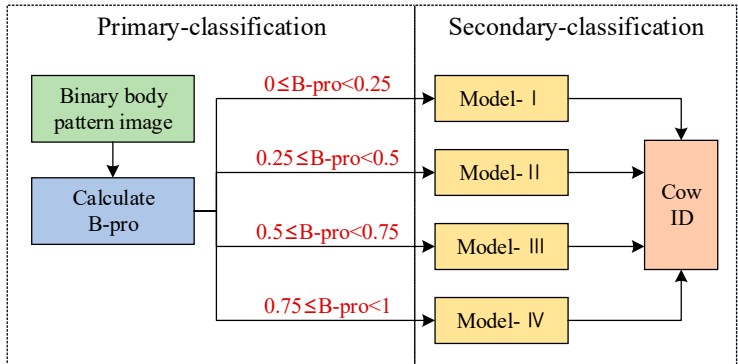

**Figure 5.** Cascaded classification model.

2.4.2. Secondary Classification

The four subcategories generated by primary classification correspond to the four different secondary classification models. According to the result of primary classification, the image was assigned to the corresponding secondary classification model for individual identification (as shown in Figure 5). Secondary classification was based on the distribution characteristics of black and white patterns in images. Because the binarization process filters out the noise unrelated to classification in the image, secondary classification is relatively simple. The network does not need to determine which features are useful information but only needs to learn and express the features related to classification, such as the distribution area and the boundary trend of the black pattern. However, because the cow is in a state of activity, the position of feature points may change for the same cow's body pattern image, which requires the classification model to have spatial invariance. Therefore, we use EfficientNet to construct the four secondary classification networks.

The basic network architecture of EfficientNet is designed by performing a neural architecture search. EfficientNet consists of three parts. The first part contains a convolution operation, normalization processing, and an activation function whose function is to adjust the number of channels of the input image and to perform preliminary feature extraction. The second part is the main feature extraction structure of EfficientNet, which contains a stack of blocks with seven different parameters. Each block includes several mobile inverted bottleneck Convolution (MBConv) block modules. The MBConv block structure is designed with inverted residuals and ResNet in mind. First, a $1 \times 1$ convolution is used to increase the dimension, then a $3 \times 3$ or $5 \times 5$ depthwise convolution is performed, and an attention mechanism about the channel is added after this structure. Finally, a $1 \times 1$ convolution is used to reduce the dimension. The output is connected with the input side to form a residual structure. This is the unique feature extraction structure of EfficientNet, which completes efficient feature extraction in the process of block stacking. The third part of the EfficientNet-B0 network is the prediction head, which contains the convolution layer, pooling layer, and fully connected layer to obtain the final classification results. EfficientNet uses compound scaling to obtain network structures with different depths, widths, and input image sizes. The basic structure of EfficientNet-B0 is shown in Figure 6. Due to the small size of the image, the secondary classification model is selected among EfficientNet-B0, EfficientNet-B1, and EfficientNet-B2, and we determine which model to use based on the training results.

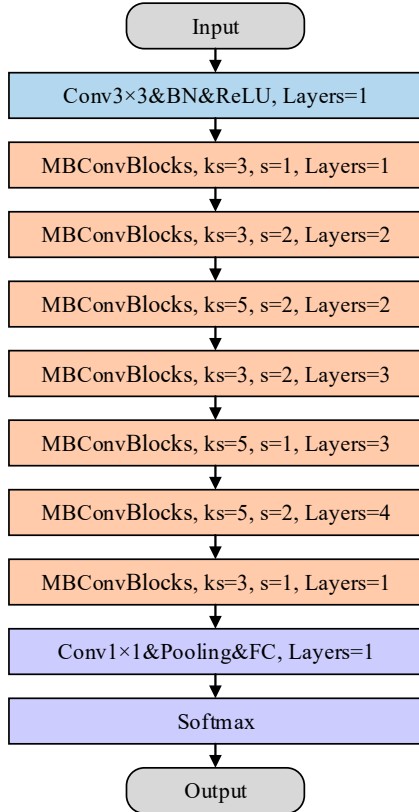

**Figure 6.** Structure of EfficientNet-B0.

### 2.4.3. Training and Testing Process

All body pattern images of each cow in the dataset were assigned to the training set, validation set, and test set at a ratio of 5:3:2 to train and test the cascaded classification model. Due to the influence of cow activity, binary segmentation error, and other factors, the proportion of black pixels in the binary body pattern image of the same cow is variable. Therefore, different binary pattern images of the same cow may be assigned to two categories in the process of primary classification. For cows in the above situation, all the

body pattern images of this cow were put into the training set of the corresponding two categories during training to ensure that no matter which category the cow is assigned to, it can be correctly identified. After primary classification, the secondary classification models corresponding to the four categories were trained based on EfficientNet-B0, EfficientNet-B1, and EfficientNet-B2. By comparing their training results, the network structure with higher accuracy was selected as the secondary classification network.

After network training and structure selection, the images in the test set were used to evaluate the performance of the cascaded classification model. The binary body pattern images in the test set were put into the primary classification model first, and then the images were transferred into the corresponding secondary classification model for individual identification. After cascaded classification was completed, the classification accuracy rate $Acc_{cls}$ was used as an index to evaluate the accuracy of the model, and the detection time of a single image was used as an index to evaluate the efficiency of the model. $Acc_{cls}$ is calculated as follows:

$$Acc_{cls} = \frac{true}{true + false} \tag{8}$$

where *true* represents the number of correctly classified samples and *false* represents the number of misclassified samples.

## 3. Results

### 3.1. Analysis of Trunk Area Detection Results

The test set in the subdataset was put into the trained YOLOX model to test the performance in cow trunk detection. The results showed that the accuracy evaluation index AP75 value of the detection model reached 0.988, the AP value reached 0.843, and the detection time of a single image was 0.023 s. The YOLOX algorithm can accurately and efficiently obtain the position of the cow trunk from the side-looking walking image of a cow. Figure 7 shows some detection results with different lighting scenes and body patterns. The figure shows that the YOLOX model has good robustness, and that the detection bounding box can contain the trunk area with body patterns, retain the main features used in individual identification, and eliminate interference in the background.

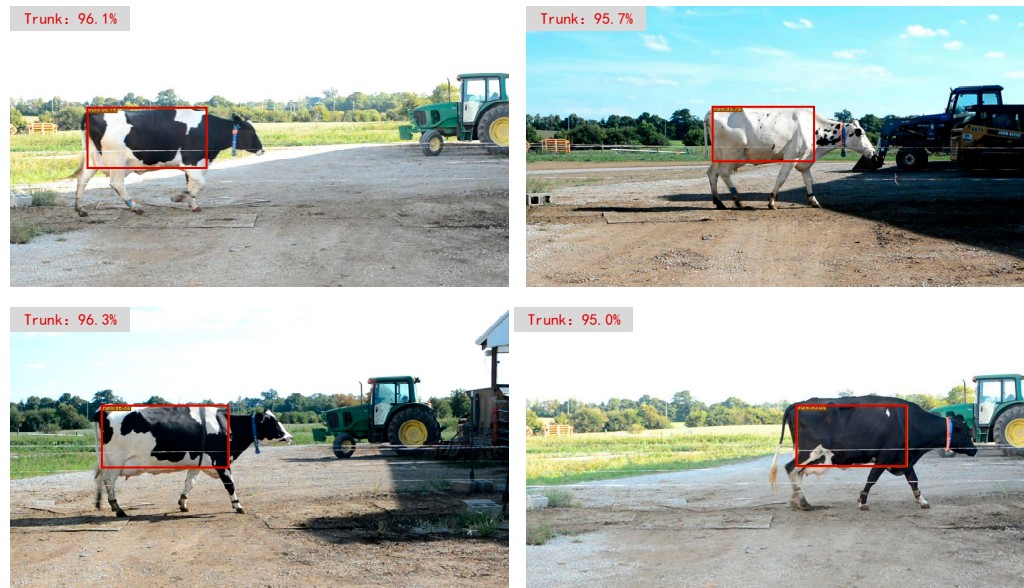

**Figure 7.** Cow trunk detection results. The red rectangle in the figure represents the detection bounding box of the trunk area, and the text in the upper left of the image represents the category and confidence of the detection bounding box.

### 3.2. Analysis of the Binarization Results of Body Pattern Images

The test set in the subdataset was put into the traditional binarization models and trained DeepOtsu model, to test the performance in cow body pattern image binarization. The test results of the three methods showed that the DeepOtsu method achieved the highest binarization accuracy of 0.932, the binarization method based on color features achieved an accuracy of 0.877, and Otsu's method based on the gray distribution achieved the lowest accuracy of only 0.827.

Figure 8 shows the binarization results of the three methods for the cow trunk images with interference. The figure shows that the grayscale conversion process reduces the redundant information of the image and filters out some useful information for binarization, resulting in bad body pattern image binarization results. The color feature, as a simple description method, cannot better describe the distribution of black and white body patterns of dairy cows. Therefore, the binarization method based on color features and gray features cannot solve the binarization problem of cow body pattern images in complex scenes. Compared with the other two methods, the DeepOtsu model has obvious advantages and has good robustness to complex interference situations. The DeepOtsu model can remove reflections, stains, shadows, and occlusions in the image through the convolutional neural network to obtain a satisfactory binary image. Therefore, this study used DeepOtsu as a binarization method for cow body pattern images.

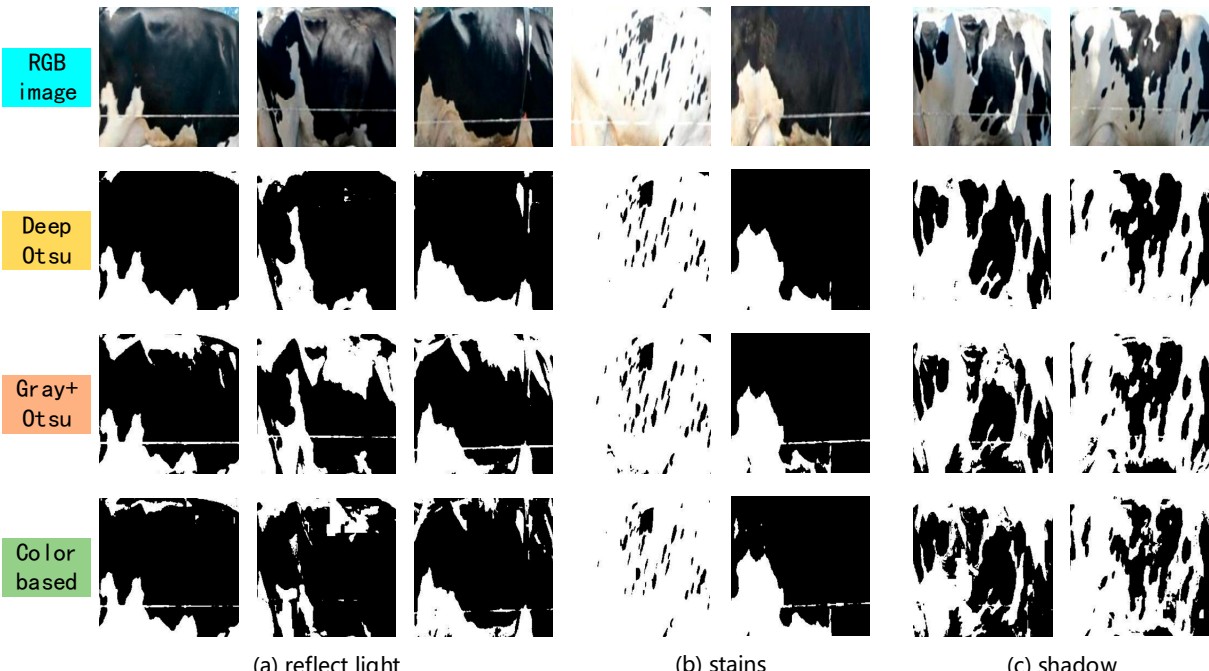

(a) reflect light   (b) stains   (c) shadow

**Figure 8.** Comparison of three binarization methods under different conditions. In the figure, the images in the first row represent the RGB images to be binarized; the images in the second row represent the images binarized by the DeepOtsu model; the images in the third row represent the images after the grayscale conversion process and Otsu binarization; and the images in the fourth row represent images processed by color-based binarization. (**a**) reflect light (**b**) stains (**c**) shadow.

Figure 9 shows the segmentation results of DeepOtsu model in the presence of interference. The main disturbances that affect the binarization accuracy of the cow body pattern image are as follows.

- The reflection of the black hair area is caused by strong light, which makes the area very bright, as shown in the red rectangle in Figure 9.
- The white electric fence used to limit the walking range of cows leaves a linear white mark on the image of cow body patterns, as shown in the green rectangle in Figure 9.

- The stain in the trunk area makes the area dark, as shown in the yellow rectangle in Figure 9.
- Bright and dark areas are formed by the shadow on the cow, as shown in the blue rectangle in Figure 9.
- Slight overexposure causes the overall image to be brighter, as shown in the last column of Figure 9.

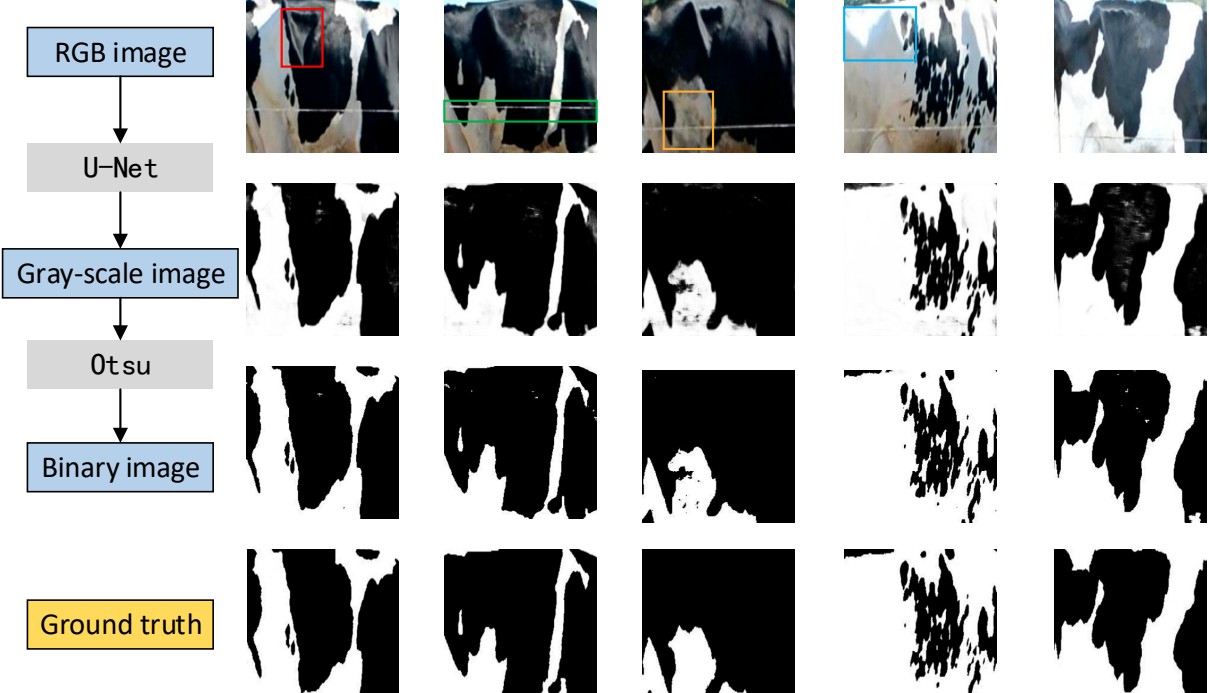

**Figure 9.** Binarization segmentation results of cow body pattern images. In the figure, the images in the first row are the RGB body pattern images to be processed; the images in the second row are the gray images after U-Net segmentation; the images in the third row are the binary images after Otsu processing; and the images in the fourth row are the ground truths for comparison. The colored rectangular box in the figure marks some areas with interference.

The segmentation results in Figure 9 show that the DeepOtsu model can eliminate different kinds of interferences in the image and output satisfactory binary images of the cow body pattern. By using the convolution neural network U-Net, a relatively 'clean' grayscale cow body pattern image was generated to obtain better binarization results. The binarization process can eliminate the redundant information in the image so that the image only contains the distribution characteristics of black and white patterns. For the individual identification model, the binarization process plays a role in improving the image quality. The binarized cow body pattern image is used as the input of the cascaded classification model, which can make the classification network learn the useful information in the image more quickly and accurately, reduce the complexity of the individual identification model, and make the model adapt to more complex and changeable scenes. Although there are still some small areas that were wrongly segmented in the image, the main features of the black and white pattern distribution were still retained. In the classification process, these misclassified small areas have little effect on the results.

### 3.3. Analysis of Individual Identification Results of Dairy Cows

3.3.1. Training Results

The proportion values of the black pixels of the binarized cow body trunk images in the training set were counted. According to the proportion values, images were assigned to four categories. The number of cows in each category is shown in Table 1. Different binary pattern images of the same cow may be assigned to two categories due to the changes in *B-pro* values. Therefore, the total number of cows in the four categories is greater than 118. The table shows that the number of cows in categories I and II is less, and the number of cows in categories III and IV is more. Figure 10 shows partial binary cow body pattern images in four categories.

**Table 1.** Primary classification results.

| Index | I | II | III | IV |
|---|---|---|---|---|
| The number of cows | 23 | 29 | 49 | 47 |

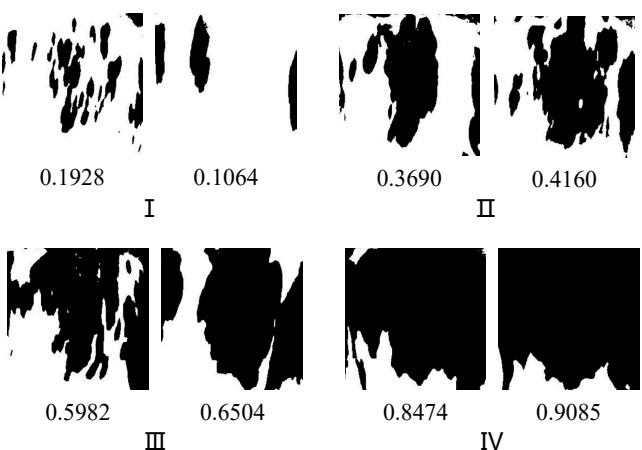

**Figure 10.** Binarized cow body pattern images in four categories. In the figure, the number below each image represents the proportion of black pixels in that image.

After primary classification was completed, the training sets of different categories were put into EfficientNet-B0, EfficientNet-B1, and EfficientNet-B2 for training. The training results of the four secondary classification models are shown in Table 2. The table shows that for the four secondary classification tasks, the training accuracy of EfficientNet-B1 is better than the training accuracies of EfficientNet-B0 and EfficientNet-B2. At the same time, the training results show that the training accuracy of EfficientNet-B2 is very poor, which may be due to the overfitting of the network caused by the small image size and small amount of data. The depth of the EfficientNet-B1 network is sufficient to extract deep features from the binary cow body pattern image, so EfficientNet-B1 was selected as the secondary classification model.

**Table 2.** Training accuracy of the four categories.

| Model | I | II | III | IV |
|---|---|---|---|---|
| EfficientNet-B0 | 1 | 1 | 0.985 | 0.963 |
| EfficientNet-B1 | 1 | 1 | 0.997 | 0.971 |
| EfficientNet-B2 | 0.372 | 0.274 | 0.125 | 0.128 |

3.3.2. Test Results

The images in the test set were put into the cascaded classification model for primary classification and secondary classification, and the classification results and the classification time of a single image were counted. According to the statistics, all the images were classified correctly in primary classification. For secondary classification, the classification results for different categories are shown in Table 3.

**Table 3.** Test results of secondary classification.

| Index | I | II | III | IV | Average |
|---|---|---|---|---|---|
| $Acc_{cls}$ | 1 | 1 | 0.991 | 0.949 | 0.985 |
| Classification time for a single image/s | 0.389 | 0.408 | 0.412 | 0.412 | 0.405 |

The table shows that the classification accuracy rate of categories I and II is 1, the classification accuracy rate of category III is the second highest, and the classification accuracy of category IV is the lowest. The number of output ends of categories I and II is relatively small. Figure 10 shows that the proportion values of black pixels in the body pattern images belonging to categories I and II are relatively low, so the distribution characteristics of black and white patterns are rich. Therefore, the accuracy of these two categories reaches 100%. The number of cows belonging to category III is almost twice that belonging to categories I and II, so the classification accuracy is slightly lower. However, the distribution features of black and white patterns in the binary speckle image are still relatively rich, so its classification accuracy is also very high. The number of cows belonging to category IV is also relatively large. Figure 10 shows that the images in category IV have a relatively high proportion of black pixels, and most of the images have large black areas. The areas with distinguishable feature points are small and generally located at the bottom or corners of the image, so the overall classification accuracy of the images in category IV is slightly low. In addition, the reflection of the black hair area is the main reason for the reduced binarization accuracy. Obviously, the cows belonging to category IV have relatively more black hair area in their body pattern and more binarization errors, which makes the corresponding classification accuracy lower. Overall, the average classification accuracy of the four secondary-classification models is 0.985, which achieved high accuracy in individual identification.

In addition, from the classification results of the four categories, the number of outputs affects the accuracy of the classification model. Reducing the number of outputs of the classification model can improve the process accuracy and speed of the individual cow identification model, and the resulting model has better recognition ability for cows with similar body patterns.

## 4. Discussion

*4.1. Comparison between the Cascaded Method and End-to-End Method*

To compare the cascaded identification method with the end-to-end identification method, all RGB body pattern images of each cow in the dataset were used to construct the training set, validation set, and test set at a ratio of 5:3:2, and the end-to-end identification model based on EfficientNet-B1 was trained. Table 4 shows the identification accuracy and speed of the end-to-end method and the cascaded method.

**Table 4.** Identification accuracies and speeds of different methods.

| Index | Cascaded Method | End-to-End Method |
|---|---|---|
| $Acc_{cls}$ | 0.985 | 0.987 |
| Identification time of a single image/s | 0.405 | 0.432 |

Table 4 shows that the end-to-end individual identification method and the cascaded individual identification method achieve the same high accuracy, which is above 0.98. However, because the cascaded individual identification method reduces the number of outputs of each secondary classification model, the number of parameters of the cascaded individual identification model is less than that of the end-to-end individual identification model, so the processing speed of the cascaded individual identification method is slightly higher than that of the end-to-end individual identification method.

In practical applications, when a new cow joins the dairy farm, the recognition model needs to be retrained. Therefore, this paper counts the training time of different individual identification methods, as shown in Table 5. For the cascaded individual identification method, only one or two secondary classification models need to be retrained when a new cow is added (in most cases, only one model needs to be retrained). For the end-to-end recognition method, the entire model needs to be retrained when a new cow is added. According to the comparison of training time in Table 5, the training time of the cascaded individual identification method is shorter than that of the end-to-end individual identification method.

**Table 5.** Training times of different individual identification methods.

| Index | Cascaded Method | | | | End-to-End Method |
| | I | II | III | IV | EfficientNet-B1 |
|---|---|---|---|---|---|
| Training time/min | 32 | 39 | 70 | 66 | 132 |

### 4.2. Error Analysis

In this paper, the statistics and analysis of the error individual identification results were carried out. Figure 11 shows the two cows with the lowest individual identification rates in the dataset, and their individual identification rates are both 0.75. After analysis, the reasons for the low identification rate include the following two aspects: (1) The two cows belong to category IV, meaning that the distribution area of black and white patterns in the trunk area is very small, and fewer corresponding identification features exist. (2) The distribution of black and white patterns is concentrated in the bottom area of the trunk, and leg movement will change the distribution and shape of the patterns when the cow walks, thus affecting the secondary classification accuracy. Because of the small number of samples in the training set, these changes cannot be learned by the secondary classification model, which is also one of the reasons for the low individual identification rate.

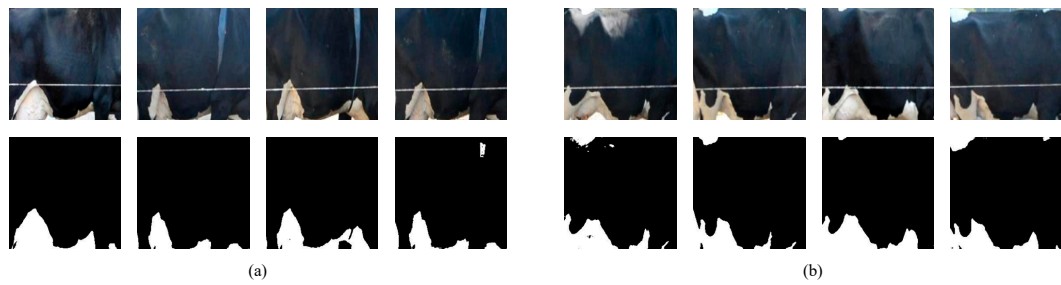

(a) (b)

**Figure 11.** Body pattern images of two cows (**a**,**b**) with the lowest identification rate.

### 4.3. Comparison of the Proposed Method with Similar Studies

In order to show the advantages of the proposed method more intuitively, a comparison with other identification method based on body pattern images [8,20,25,26,33,34] was conducted as illustrated in Table 6. It can be seen from Table 6 that the number of cows in the dataset of this paper is the largest, and the identification accuracy of the proposed method exceeds most of the references in the table. Although the accuracy in [20,33] is higher than our proposed method in this paper, the dataset of [20] contains only 10 cows,

and the number of cows is very small. It is needed to collect cow images from four perspectives in [33], so the time and labor cost of collecting data are high. In summary, the cascaded individual cow identification method proposed in this paper has obvious advantages over the other publishing similar research and has the potential to be applied to large-scale automated pastures.

**Table 6.** Comparison between our proposed method and some of state-of-the-art methods in term of image source, identification accuracy, and number of cows in the datasets.

| Reference | Image Source | Identification Accuracy | Number of Cows |
| --- | --- | --- | --- |
| [8] | Side view images of cow | 98.36% | 93 |
| [20] | Tailhead images | 99.7% | 10 |
| [25] | Back images of cow | 95.91% | 89 |
| [33] | Back image, left side profile image, right side profile image, facial image | 99% | 51 |
| [26] | Side view images of cow | 96.65 | 105 |
| [34] | Body pattern images (top view) | 93.8 | 46 |
| Our method | Body pattern images (side view) | 98.5 | 118 |

*4.4. Future Research*

Although our proposed cascaded method can achieve fast and accurate individual identification of dairy cows, there is still room for improvement. For the binary segmentation of cow trunk images, severely overexposed images were removed when constructing the dataset. However, in an actual production environment, overexposure occasionally occurs. Therefore, in future studies, we can improve the robustness of the binarization model by optimizing the algorithm network so that the cascaded dairy individual cow identification method can adapt to more complex scenes on farms. In addition, due to the limitation of data collection conditions, the number of cows in the dataset constructed in this paper is relatively small, and the number of samples per cow is also relatively small. In future studies, the data can be collected on a large-scale dairy farm with more cows. The proposed method can be applied to farms to further verify the superiority of the method compared with the end-to-end identification method and its potential application on large-scale dairy farms.

**5. Conclusions**

In this paper, a method of cascaded individual dairy cow identification based on DeepOtsu and EfficientNet was proposed. The body pattern images of dairy cows were binarized and cascaded classified to address the identification difficulty caused by similar body pattern characteristics, poor image quality, and a large number of output terminals in dairy cow group identification. The test results of the method showed that the detection accuracy (AP75) of the cow trunk based on YOLOX is 0.988, and the detection time of a single image is 0.023 s; the binarization accuracy of cow body pattern images based on DeepOtsu is 0.932, and the binarization time of a single image is 0.005 s. The classification accuracy of the cascaded classification model is 0.985, and the classification time of a single image is 0.405 s. The overall individual identification accuracy of the proposed method is 0.985, and the identification time of a single image is 0.433 s. Compared with the end-to-end individual identification method, the proposed method has obvious advantages in identification efficiency and training speed. The proposed method provides a new approach to dairy cattle group individual identification in large-scale dairy farms.

**Author Contributions:** Conceptualization, R.Z., J.J. and K.Z.; Methodology, R.Z., K.Z. and M.Z.; Software, R.Z., J.W. and M.Z.; Validation, R.Z.; Formal analysis, R.Z. and J.J.; Investigation, K.Z.; Resources, R.Z., J.J. and K.Z.; Data curation, K.Z. and J.W.; Writing—original draft, R.Z.; Writing—review and editing, K.Z. and M.W.; Visualization, R.Z. and J.W.; Supervision, J.J.; Project administration, K.Z.; Funding acquisition, J.J. and M.W. All authors have read and agreed to the published version of the manuscript.

**Funding:** This research was funded by "National Key R&D Plan Key projects of Scientific and technological Innovation Cooperation between Governments", grant number "2019YFE0125600"; "National Natural Science Foundation of China", grant number 32002227; and "Natural Science Basic Research Plan in Shaanxi Province of China", grant number "2022JQ-175".

**Institutional Review Board Statement:** Not applicable.

**Data Availability Statement:** Not applicable.

**Acknowledgments:** The authors acknowledge University of Kentucky for facilitation of data acquisition and permission for data use. This study was supported by the National Natural Science Foundation of China (grant No. 32002227), the National Key R&D Plan Key projects of Scientific and technological Innovation Cooperation between Governments (grant No. 2019YFE0125600), and the Natural Science Basic Research Plan in Shaanxi Province of China (grant No. 2022JQ-175).

**Conflicts of Interest:** The authors declare no conflict of interest.

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
