# Peer review of "A Cascaded Individual Cow Identification Method Based on DeepOtsu and EfficientNet"

_agriculture, doi:10.3390/agriculture13020279_

Round 1

Reviewer 1 Report

Dear Authors,

The article is very interesting and seems to be balanced, well-structured and written in readable English. However, I have some suggestions which can help you to improve the work, please find them below:

1.               The indicators AP and AP75 in line 288 should be defined.

2.               Please, consider the replacement of the section 4.1. I think it should be before results of DeepOtsu because this part of the research should be base of the decision of the use of DeepOtsu in the final model.

3.               The “Discussion” paragraph is in fact additional research. I suggest to the Authors to add the subsection with classical discussion with results obtained by others publishing similar research before the subsection 4.4. Future Research

Author Response

Response to Reviewer 1 Comments

Dear reviewer:

Thanks for the comments on our manuscript. We have made the responses (red font) point by point, and the corresponding revisions can be found in track-change manuscript.

Point 1: The indicators AP and AP75 in line 288 should be defined.

Response 1: The definitions of indicators AP and AP75 have been added in the fourth paragraph of section 2.2. Please check the revised paper for details.

Point 2: Please, consider the replacement of the section 4.1. I think it should be before results of DeepOtsu because this part of the research should be base of the decision of the use of DeepOtsu in the final model.

Response 2: This part was separated and the description of traditional binarization methods was placed in section 2.3.1. The comparison results of different binarization methods were placed in section 3.2. Please check the revised paper for details.

Point 3: The “Discussion” paragraph is in fact additional research. I suggest to the Authors to add the subsection with classical discussion with results obtained by others publishing similar research before the subsection 4.4. Future Research.

Response 3: The classical discussion with results obtained by other publishing similar research has been added in section 4.3 Comparison of the proposed method with similar studies. Please check the revised paper for details.

Reviewer 2 Report

The article deals with another utilization of various organized NNs, in sinergy, intended for objects classification and segmentation and applied in an agricultural case of live stock localization and recognition. A focus is set on emphasis and justification of NN employement for ground and validation set images binarization and proper diversification, furtherly classified accordingly.

Despite article's interesting application and correspondingly produced results some aspects have to be improved, to even more rise its quality and soundness. The aspets that have to be improved are:

- Explain in more detailed way a decision of YoloX choice vs. Yolo5 (and 3). A sole statement that '...Yolox's performane exceed those of Yolo5 and Yolo3...' isn't well supportive. Please, elaborate in quantitative manner.

- How did you avoid overfitting issues with such modest set of training and validating images?

- How does, in your case, background noise specifficaly influences on a binarization results? Sentence or two of concise explanation will suffice.

Author Response

Response to Reviewer 2 Comments

Dear reviewer:

Thanks for the comments on our manuscript. We have made the responses (red font) point by point, and the corresponding revisions can be found in track-change manuscript.

Point 1: Explain in more detailed way a decision of YoloX choice vs. Yolo5 (and 3). A sole statement that '...Yolox's performane exceed those of Yolo5 and Yolo3...' isn't well supportive. Please, elaborate in quantitative manner.

Response 1: At the end of the first paragraph of section 2.2, the advantages of YOLOX in terms of precision compared with YOLO series networks have been quantitatively elaborated: “YOLOX achieves 50.0% AP on COCO (1.8% higher than YOLOv5 and 2.5% higher than YOLOv4)[31]. And the precision of YOLOX is much higher than that of YOLOv3 (33.0% AP). Therefore, we finally decided to use YOLOX to detect the trunk area of dairy cows.” Please check the revised paper for details.

Point 2: How did you avoid overfitting issues with such modest set of training and validating images?

Response 2: According to our experience and literature [1], for modest datasets, choosing a simpler network structure can effectively avoid the problem of overfitting. This is also reflected in the Table 2. Training accuracy of the four categories in the revised paper. The network structure of EfficientNet-B2 is deeper, wider and more complex than EfficientNet-B0 and EfficientNet-B1, but its classification accuracy is very low, indicating that there is an over-fitting problem in the training process of EfficientNet-B2.

Point 3: How does, in your case, background noise specifficaly influences on a binarization results? Sentence or two of concise explanation will suffice.

Response 3: “There were noises from light, stains, occlusion in the background of the cow trunk images, which will lead to wrong binarization results. For example, the reflection caused by strong light makes the black hair area very bright, then the binarization result of this black hair area is easily misclassified as white (value 1); the presence of stains in the white hair area will cause the area to darken, and its binarization result is easily misclassified as black (value 0).” The above content was added to the first paragraph of section 2.3.2.

  1. He, Z., Xie, Z., Zhu, Q., Qin, Z. Sparse Double Descent: Where Network Pruning Aggravates Overfitting. In International Conference on Machine Learning (pp. 8635-8659). PMLR, Baltimore (June 2022).
